# Morphological Comparison of the Chesapeake Logperch *Percina bimaculata* with the Logperch *Percina c. caprodes* and *Percina c. semifasciata* in Pennsylvania

**Jay R. Stauffer, Jr. [1,2,\*], Jonathan A. Freedman [3,†], Douglas P. Fischer [4] and Robert W. Criswell [5,‡]**

1 Ecosystem Science and Management, Penn State University, 432 Forest Resources Building, University Park, PA 16802, USA
2 South African Institute for Aquatic Biodiversity, Makhanda 6140, South Africa
3 School of Forest Resources, Penn State University, University Park, PA 16802, USA; freedjon@gmail.com
4 Pennsylvania Fish and Boat Commission, 595 Rolling Ridge Drive, Bellefonte, PA 16823, USA; doufischer@pa.gov
5 Pennsylvania Game Commission, Harrisburg, PA 19019, USA; rob_criswell@comcast.net
\* Correspondence: vc5@psu.edu; Tel.: +1-814-404-1662
† Current address: Cherokee Nation System Solutions, Contracted to U.S. Geological Survey, Nonindigenous Aquatic Species Program, Wetland and Aquatic Research Center, Gainesville, FL 32605, USA.
‡ This person is retired.

**Abstract:** The Chesapeake logperch, *Percina bimaculate* (Halderman) has a disjunct distribution when compared to other species in the subgenus *Percina*. Members of this subgenus in Pennsylvania include *Percina caprodes caprodes* (Rafinesque), *Percina caprodes semifasciata* (DeKay), and *P. bimaculata*. Historically the Chesapeake logperch was known only from the Susquehanna River and Potomac River basins. Its range is now restricted to the Susquehanna River below Holtwood Dam and upper Chesapeake Bay. It has been extirpated from the Potomac River and the type locality near Columbia, PA. Attempts are being made to reintroduce it into tributaries of the Susquehanna River near Columbia, PA. We postulate that *P. bimaculata* diverged from a population of *Percina caprodes semifasciata*. A naked nape and the fact that both of these species do not occur above the fall line in Pennsylvania support such a relationship.

**Keywords:** *Percina bimaculata*; zoogeography; taxonomic relationships

**Key Contribution:** Although *P. bimaculata* and *P. c. semifasciata* are quite different morphologically, we postulate that *P. bimaculata* diverged from a population of *P. c. semifasciata* that originated in the Great Lakes and occupied the Potomac and Susquehanna rivers.

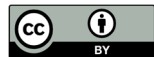

## 1. Introduction

The Chesapeake logperch, *Percina bimaculata*, (Percidae) was described by Haldeman [1] from the Susquehanna River, near Columbia, Lancaster County, PA as *Perca* (*Percina*) *nebulosa*. In 1844, Haldeman [2] described *P. bimaculata* from the same area of the river. The two were subsequently deemed to be the same species, and were eventually synonymized with the Logperch, *Percina caprodes* (Rafinesque), which ranges through the Great Lakes-St. Lawrence, Hudson Bay, and Mississippi River systems [3]. *Percina c. caprodes* was originally described from the Ohio River. Subsequently, DeKaye [4] described *Percina c. semifasciata* from Lake Champlain, NY and Morris and Page [5] described *Percina c. fulvitaenia* from the Big Piney River (Gasconade drainage) from Texas County, Missouri. Currently, the Logperch is comprised of three subspecies: *P. c. caprodes* (Rafinesque), *P. c. semifasciata* [4], and *P. c. fulvitaenia* [5]. Near [6], using morphological characters and mitochondrial and nuclear gene sequence data, concluded that the

Chesapeake logperch is in fact a distinct species and that it was closely related to the Mobile logperch, *Percina kathae*, and the Southern logperch, *Percina austroperca*. The epithet *Perca nebulosa* was already occupied; thus, *P. bimaculata* (Haldeman) became the valid scientific name [6].

Historically, the Chesapeake logperch was known from the Potomac River including collections in the District of Columbia, Maryland, and Virginia [7]. It was known from tributaries of the Potomac south of Washington D.C., the Chesapeake and Ohio Canal and Pumunkey Creek, D.C. It is currently considered extirpated from the Potomac River drainage [8,9]. Although once recorded as far upstream as Columbia, PA, in the Susquehanna River [1,2], after 2019, it was only known from the Susquehanna River and its tributaries downstream of Holtwood Dam, PA to just below Havre de Grace, MD. Efforts by the Pennsylvania Fish and Boat Commission have attempted to reintroduce it to the Conewago and Chiques creeks in Pennsylvania starting in 2019. In addition to the Chesapeake logperch, two subspecies of *Percina caprodes* (Rafinesque), Ohio Logperch, *P. c. caprodes*, and Northern Logperch, *P. c. semifasciata*, are known from Pennsylvania [10]. The purpose of this paper is to morphologically compare populations of this subgenus that inhabit Maryland and Pennsylvania, report on the current distribution, and discuss possible origins of the Chesapeake logperch.

## 2. Materials and Methods

We collected all of the fish, except those examined from the Illinois Natural History Survey, using seines, back-pack electrofishing units, boat electrofishing, or electrified benthic trawls [11]. All of the fish were anesthetized with clove oil, euthanized in 1% formalin, preserved in 10% formalin, and placed in permanent storage in 70% ethanol in the Pennsylvania State University Fish Museum. All collections followed the methods approved by the Animal Use and Care Committee at Pennsylvania State University (PROTP201800659).

Eighteen measurements and seven counts were taken for each individual following the procedure of Konings and Stauffer [12]. The morphometric data were taken with digital calipers and measured to the nearest 0.1 mm. All rays of the pectoral fin were counted, including the small splinter on the upper edge of the fin. Lateral-line scales were counted from the anterior to the hypural plate. Pored lateral-line scales posterior to the hypural plate were counted separately. All counts and measurements were taken from the left side of the body except for gill-raker counts, which were taken on the right side.

*Percina bimaculata* were collected from tributaries of the Susquehanna River below Holtwood Dam (i.e., Muddy Creek (DPF-19-014), Peters Creek (DPF-19-045; KHC-19-04), Fishing Creek (DPF-19-011, RWC-05-041), Michael Creek (KHC-19-17)), the West Branch of Octoraro Creek (DPF-19-050, KHC-19-05), and the East Branch of Octoraro Creek in Pennsylvania. In Maryland, they were collected from just below Conowingo Dam (DPF-19-19, DPF-19-24, DPF-19-25). *Percina c. semifasciata* were collected from Presque Isle Bay (PSU 2348), Walnut Creek (PSU 1628), 16-Mile Creek (JAF-09-02), and Crooked Creek (JAF-09-06) in Pennsylvania and the Genesee River (PSU 4938) at Wellsville in New York. Additional specimens were from the Mississippi River drainage in Illinois (INHS 65009, 3310, 45338, 92649, 91919, 37316, 4065). *Percina c. caprodes* were collected from the Allegheny River (BDL-08-213,203,08; BDL-09-16, JAF-08-06, 07), French Creek (PSU 1832, 2240), and Raccoon Creek (PSU 1618) in Pennsylvania, from the West Fork River, Weston, WV (PSU 1817), and from the Allegheny River near Tunungwant Creek, NY (PSU 4799) (Figure 1).

Analysis of morphometrics and meristics was conducted using sheared principal component analysis (SPCA) and principal component analysis (PCA), respectively, as described by Humphries et al. [13] and Stauffer et al. [14]. Principal component analysis was used to analyze meristic data with the correlation matrix factored. Body shape differences were analyzed using SPCA with the covariance matrix factored. To illustrate differences in the counts and measurements among the species, the sheared second

principal components of the morphometric data were plotted against the first principal components of the meristic data. The first sheared principal component of the morphometric data accounted for variation in individual size. Similarly, the sheared second principal components explained the additional variation in shape. An ANOVA, in conjunction with Duncan's multiple range test ($p < 0.05$) was used to determine if the populations differed along the axis of the sheared second principal component (morphometric data) or the axis of the first principal component of the meristic data. If the clusters were not significantly different along one axis independent of the other, then a MANOVA, in conjunction with a Hotelling–Lawley trace, was used to determine whether the mean multivariate scores of the clusters formed by the minimum polygons of the PCA scores were significantly different ($p < 0.05$).

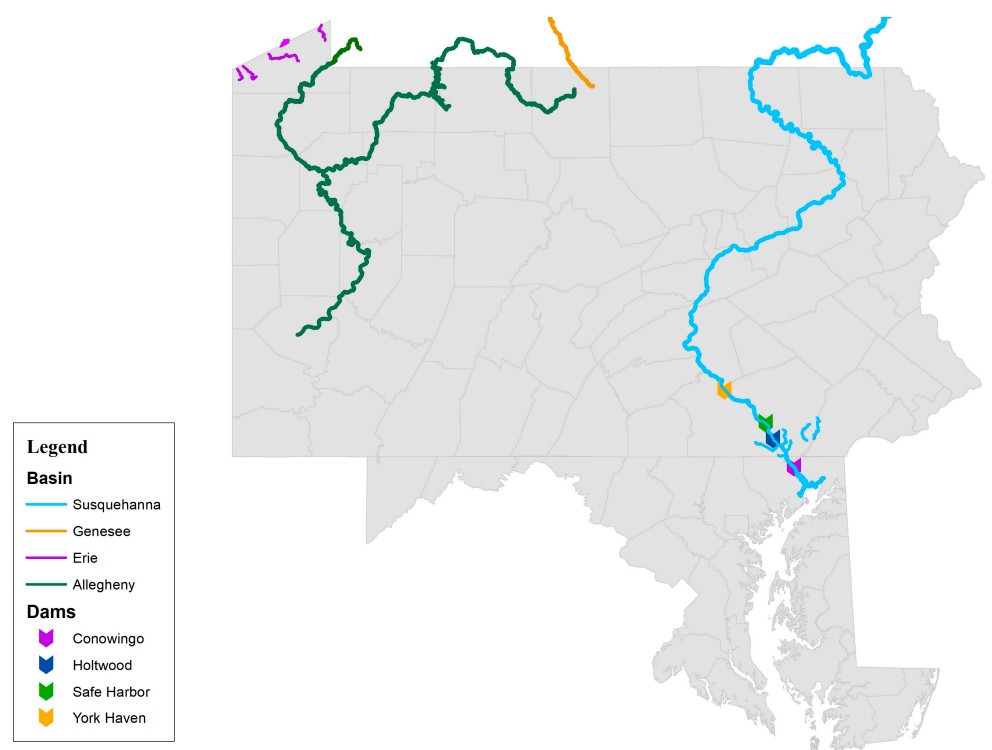

**Figure 1.** Map showing drainages in which the fish were collected.

## 3. Results

The morphometric and meristic data for *P. bimaculata* captured in Octoraro Creek and the Susquehanna River are summarized in Table 1. A plot of the SHRD PC2 of the morphometric data against the PC1 of the meristic data illustrates that the minimum polygon clusters are coincident (Figure 2). A MANOVA showed that these clusters were not significantly different ($p > 0.05$).

**Table 1.** Morphometric and meristic data for *Percina bimaculata*.

|  | Octoraro Creek (n = 19) | | | Susquehanna River (n = 27) | | |
|---|---|---|---|---|---|---|
| **Variable** | **Mean** | **Min** | **Max** | **Mean** | **Min** | **Max** |
| Standard length | 80.8 | 55.3 | 166.2 | 69.5 | 51.2 | 90.0 |
| Head length | 21.2 | 15.4 | 29.5 | 18.2 | 13.7 | 23.8 |
| **Percent head length (%)** | | | | | | |
| Snout length | 33.7 | 30.4 | 37.1 | 32.7 | 29.4 | 36.5 |
| Postorbital head length | 47.4 | 43.7 | 49.1 | 47.3 | 43.0 | 50.6 |

| | | | | | | |
|---|---|---|---|---|---|---|
| Horizontal eye diameter | 21.5 | 18.9 | 24.3 | 22.9 | 20.3 | 24.7 |
| Vertical eye diameter | 21.2 | 18.6 | 24.7 | 22.4 | 19.5 | 25.0 |
| Head depth | 53.4 | 49.3 | 58.6 | 53.5 | 45.8 | 64.1 |
| **Percent standard length (%)** | | | | | | |
| Head length | 26.3 | 24.9 | 28.1 | 26.3 | 24.6 | 28.3 |
| Snout to first dorsal fin origin | 32.5 | 30.8 | 34.1 | 32.8 | 31.8 | 34.7 |
| Snout to pelvic fin origin | 30.8 | 28.8 | 33.4 | 20.2 | 27.7 | 33.1 |
| First dorsal fin base length | 31.4 | 29.2 | 33.8 | 31.4 | 29.2 | 33.2 |
| Second dorsal fin base length | 21.7 | 19.7 | 23.0 | 21.5 | 19.5 | 23.5 |
| Ant. dorsal fin to ant. anal fin | 35.3 | 33.3 | 38.0 | 35.3 | 32.0 | 37.4 |
| Post. second dorsal to ventral caudal fin | 32.8 | 31.4 | 34.4 | 32.4 | 30.4 | 34.3 |
| Posterior anal fin to dorsal caudal | 17.7 | 16.2 | 19.7 | 17.6 | 15.5 | 19.1 |
| Posterior dorsal fin to pelvic fin origin | 37.6 | 34.4 | 43.1 | 37.3 | 32.3 | 39.3 |
| Caudal peduncle length | 20.1 | 18.0 | 21.7 | 20.6 | 18.5 | 24.0 |
| Least caudal peduncle depth | 8.7 | 8.0 | 9.6 | 8.7 | 7.9 | 9.7 |
| **Meristics** | **Mode** | **Min** | **Max** | **Mode** | **Min** | **Max** |
| Dorsal fin spines | 13 | 12 | 15 | 13/14 | 13 | 15 |
| Dorsal fin rays | 15 | 14 | 16 | 15 | 14 | 16 |
| Anal fin rays | 11 | 10 | 12 | 11 | 9 | 12 |
| Pectoral fin rays | 14 | 13 | 14 | 14 | 13 | 15 |
| Pelvic fin rays | 6 | 6 | 7 | 6 | 6 | 7 |
| Lateral line scales | 75/76 | 70 | 82 | 76 | 67 | 80 |
| Pored scales posterior to the lateral line | 0 | 0 | 1 | 0 | 0 | 2 |

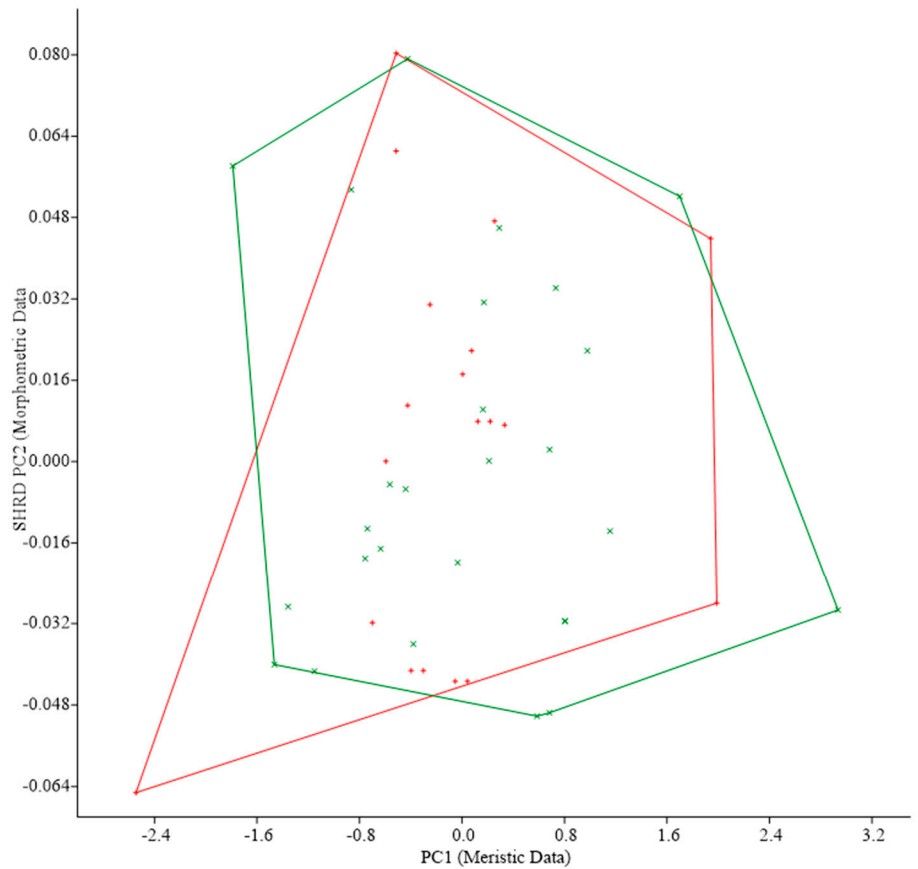

**Figure 2.** Plot of the sheared second principal components (morphometric data) and the first principal components of the meristic data for *Percina bimaculata* from Octoraro creek (+) and the Susquehanna River (x).

The morphometric and meristic data for *P. c. semifasciata* captured in Lake Erie drainage in Pennsylvania and the Mississippi River drainage in Illinois and Minnesota are summarized in Table 2, and *P. c. caprodes* from the Allegheny River and Genesee River are summarized in Table 3.

**Table 2.** Morphometric and meristic data for *Percina c. semifasciata* from the Lake Erie and Mississippi River drainage.

| | Lake Erie Drainage (n = 21) | | | Mississippi River Drainage (n = 7) | | |
|---|---|---|---|---|---|---|
| **Variable** | **Mean** | **Min** | **Max** | **Mean** | **Min** | **Max** |
| Standard length | 94.7 | 72.2 | 120.6 | 95.8 | 80.9 | 115.4 |
| Head length | 25.0 | 19.0 | 32.1 | 24.3 | 21.3 | 27.5 |
| **Percent head length (%)** | | | | | | |
| Snout length | 33.0 | 29.4 | 35.5 | 32.3 | 30.5 | 35.3 |
| Postorbital head length | 46.7 | 44.3 | 49.0 | 48.3 | 47.0 | 49.2 |
| Horizontal eye diameter | 22.3 | 20.3 | 25.4 | 22.7 | 20.3 | 23.9 |
| Vertical eye diameter | 21.1 | 18.7 | 24.0 | 22.0 | 21.0 | 22.7 |
| Head depth | 51.6 | 44.7 | 57.3 | 52.9 | 49.6 | 56.7 |
| **Percent standard length (%)** | | | | | | |
| Head length | 26.4 | 24.9 | 27.6 | 25.5 | 23.8 | 26.3 |
| Snout to first dorsal fin origin | 32.3 | 30.9 | 33.8 | 31.3 | 29.3 | 33.0 |
| Snout to pelvic fin origin | 30.8 | 27.0 | 34.4 | 29.9 | 27.9 | 31.6 |
| First dorsal fin base length | 29.7 | 26.2 | 32.3 | 31.4 | 30.2 | 32.7 |
| Second dorsal fin base length | 20.5 | 17.7 | 22.8 | 22.0 | 20.8 | 23.4 |
| Ant. dorsal fin to ant. anal fin | 36.5 | 33.5 | 39.1 | 36.4 | 34.4 | 39.0 |
| Post. second dorsal to ventral caudal fin | 18.4 | 16.7 | 19.8 | 19.0 | 17.2 | 20.5 |
| Posterior anal fin to dorsal caudal | 23.9 | 21.6 | 25.9 | 24.1 | 23.2 | 25.0 |
| Posterior dorsal fin to pelvic fin origin | 55.4 | 53.3 | 60.0 | 55.6 | 53.1 | 58.1 |
| Caudal peduncle length | 22.8 | 17.8 | 24.6 | 22.4 | 21.6 | 23.6 |
| Least caudal peduncle depth | 7.7 | 7.1 | 9.2 | 7.6 | 7.1 | 8.5 |
| **Meristics** | **Mode** | **Min** | **Max** | **Mode** | **Min** | **Max** |
| Dorsal fin spines | 14 | 13 | 15 | 14/15 | 13 | 15 |
| Dorsal fin rays | 14 | 15 | 17 | 15 | 15 | 16 |
| Anal fin rays | 7 | 7 | 8 | 6 | 6 | 7 |
| Pectoral fin rays | 12 | 10 | 13 | 12 | 11 | 14 |
| Pelvic fin rays | 7 | 6 | 7 | 6 | 6 | 7 |
| Lateral line scales | 80 | 75 | 85 | 79 | 77 | 87 |
| Pored scales posterior to the lateral line | 2 | 1 | 3 | 3 | 2 | 5 |

**Table 3.** Morphometric and meristic data for *Percina c. caprodes* from the Allegheny River drainage, PA, and Genessee River, Wellsville, NY.

| | Allegheny River (n = 44) | | | Genesee River (n = 7) | | |
|---|---|---|---|---|---|---|
| **Variable** | **Mean** | **Min** | **Max** | **Mean** | **Min** | **Max** |
| Standard length | 99.6 | 72.0 | 126.3 | 93.7 | 91.6 | 96.0 |
| Head length | 23.4 | 18.5 | 32.0 | 23.8 | 22.8 | 24.9 |
| **Percent head length (%)** | | | | | | |
| Snout length | 33.2 | 28.0 | 37.1 | 34.9 | 32.8 | 37.2 |
| Postorbital head length | 47.1 | 41.2 | 51.2 | 47.2 | 45.4 | 48.5 |

| | | | | | | |
|---|---|---|---|---|---|---|
| Horizontal eye diameter | 22.3 | 19.2 | 26.7 | 20.2 | 19.4 | 21.1 |
| Vertical eye diameter | 21.1 | 18.9 | 24.6 | 21.2 | 18.8 | 24.8 |
| Head depth | 49.2 | 38.7 | 56.5 | 49.1 | 44.3 | 52.2 |
| **Percent standard length (%)** | | | | | | |
| Head length | 25.9 | 24.3 | 28.0 | 25.3 | 24.3 | 27.1 |
| Snout to first dorsal fin origin | 31.1 | 28.6 | 34.5 | 31.7 | 30.8 | 32.4 |
| Snout to pelvic-fin origin | 29.8 | 27.4 | 33.2 | 28.0 | 26.9 | 28.8 |
| First dorsal fin base length | 31.4 | 28.6 | 34.5 | 31.9 | 30.9 | 33.3 |
| Second dorsal fin base length | 22.5 | 18.8 | 24.7 | 23.9 | 23.0 | 24.9 |
| Ant. dorsal fin to ant. anal fin | 36.0 | 32.5 | 39.3 | 35.5 | 33.9 | 37.7 |
| Post. second dorsal to ventral caudal fin | 23.6 | 20.4 | 26.2 | 17.8 | 16.8 | 18.7 |
| Posterior anal fin to dorsal caudal | 18.2 | 14.5 | 19.9 | 24.1 | 23.5 | 24.7 |
| Posterior dorsal fin to pelvic fin origin | 55.8 | 50.0 | 59.7 | 57.3 | 55.7 | 58.3 |
| Caudal peduncle length | 22.5 | 18.5 | 24.9 | 23.3 | 21.4 | 24.7 |
| Least caudal peduncle depth | 7.6 | 6.9 | 8.5 | 7.3 | 6.9 | 7.8 |
| **Meristics** | **Mode** | **Min** | **Max** | **Mode** | **Min** | **Max** |
| Dorsal fin spines | 15 | 13 | 16 | 14 | 13 | 16 |
| Dorsal fin rays | 15 | 14 | 17 | 15 | 14 | 17 |
| Anal fin rays | 7 | 6 | 7 | 7 | 7 | 7 |
| Pectoral fin rays | 13 | 11 | 13 | 12 | 11 | 13 |
| Pelvic fin rays | 7 | 6 | 7 | 7 | 7 | 7 |
| Lateral line scales | 87 | 81 | 90 | 86 | 85 | 89 |
| Pored scales posterior to the lateral line | 2 | 0 | 3 | 2 | 1 | 2 |

A plot of the SHRD PC2 of the morphometric data against the PC1 of the meristic data illustrates that the minimum polygon clusters for *P. bimaculata* do not overlap with the minimum polygon clusters of either *P. c. caprodes* or *P. c. semifasciata* (Figure 3). Size accounted for 84.3% and the second principal component accounted for 11.0% of the observed variance in the morphometric data. Variables with the highest loadings on the SPC2 components were the distance between the posterior insertion of the second dorsal fin and the insertion of the ventral caudal fin (0.76), the distance between the posterior insertion of the second dorsal fin and the insertion of the pelvic fin (−0.38), and the distance between the posterior insertion of the anal fin and the dorsal insertion of the caudal fin (−0.30). The first principal component of the meristic data accounted for 48% of the total variance. Variables with the highest loadings on the first principal component were pored scales posterior to the hypural plate (0.47), lateral line scales (0.47), and the number of anal fin rays (−0.46).

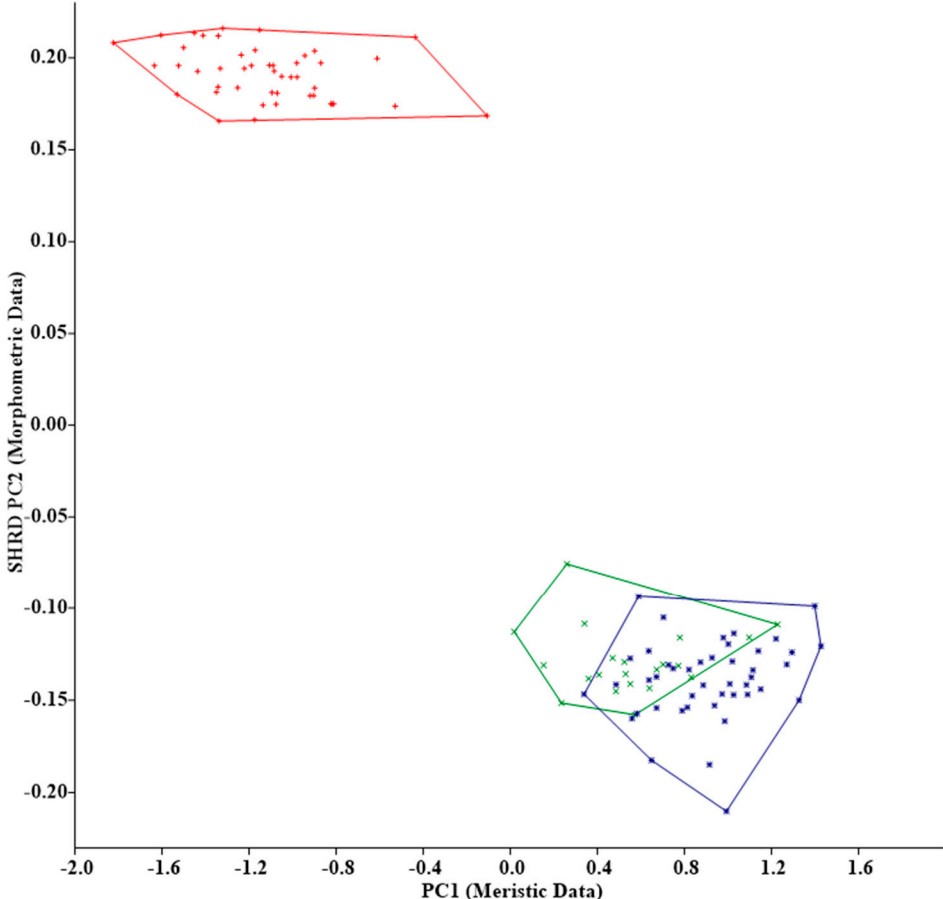

**Figure 3.** Plot of the sheared second principal components (morphometric data) and the first principal component of the meristic data for *Percina bimaculata* from PA (+), *Percina c. semifasciata* (×) from Lake Erie, and *Percina c. caprodes* (*) from the Allegheny River.

A plot of the SHRD PC2 of the morphometric data against the PC1 of the meristic data for *P. c. semifasciata* from Lake Erie, *P. c. semifasciata* from the Mississippi River, *P. c. caprodes* from the Allegheny River, and *P. c. caprodes* from the Genesee River is illustrated in Figure 3. The minimum polygon clusters were all significantly different ($p < 0.05$) from each other along the SHRD PC2 axis. The populations from the Genesee River and the Allegheny River were not significantly ($p > 0.05$) different from each other along the PC1 axis of the meristic data (Figure 4). Moreover, the populations from Lake Erie and the Mississippi River were not significantly different along the PC1 axis. The populations from the Genesee River and Allegheny River were significantly different ($p < 0.05$) from the populations from Lake Erie and the Mississippi River along PC1 (meristic data) (Figure 4). Size accounted for 85.9% and the second principal component accounted for 3.5% of the observed variance in the morphometric data. Variables with the highest loadings on the SPC2 components were the dorsal fin base length of the second dorsal fin (0.69), the dorsal fin base length of the first dorsal fin (0.33), and the horizontal eye diameter (−0.28). The first principal component of the meristic data accounted for 33% of the total variance. Variables with the highest loadings on the first principal component were lateral line scales (0.65), the number of dorsal fin spines (0.54), and the number of pectoral fin rays (0.47). The nape of *P. c. caprodes* is fully scaled, while the napes of *P. c. semifasciata* and *P. bimaculata* are naked (Figure 5).

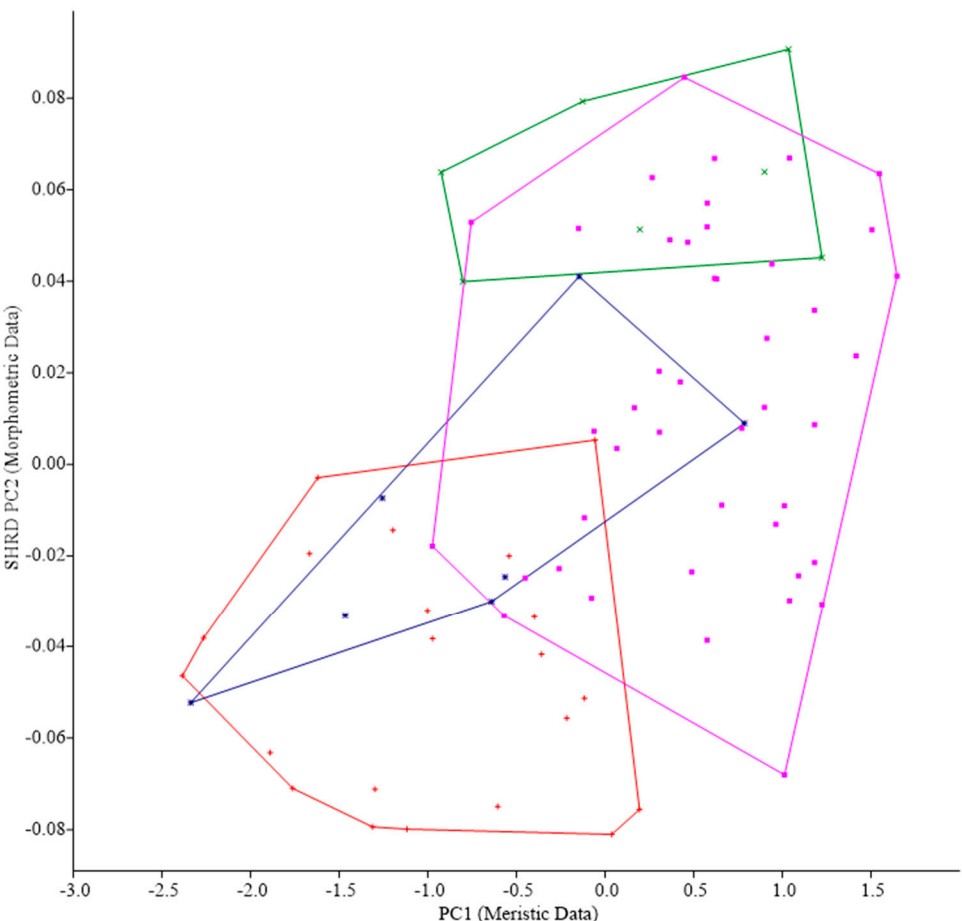

**Figure 4.** Plot of the sheared second principal components (morphometric data) and the first principal component of the meristic data for *Percina c. semifasciata* from Lake Erie (+), *Percina c. semifasciata* from the Mississippi River (*), *Percina c. caprodes* from the Allegheny River (▪), and *Percina c. caprodes* from the Genesee River (×).

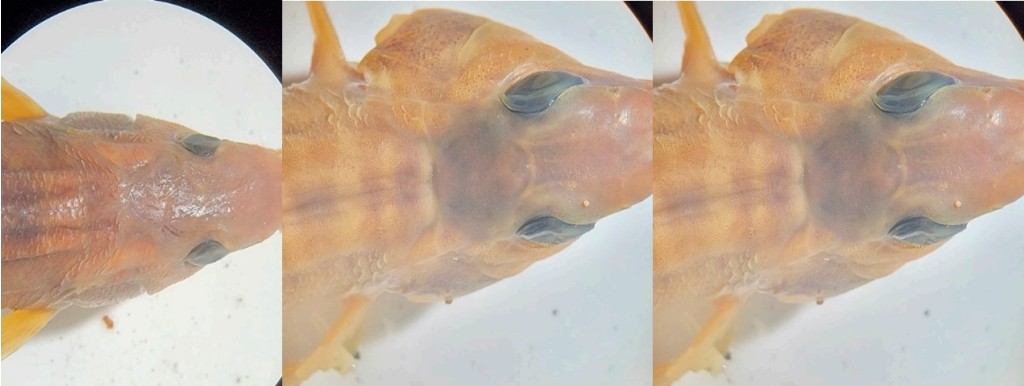

**Figure 5.** Nape of *Percina c. caprodes* (**left**), *Percina c. semifasciata* (**middle**), and *Percina bimaculata* (**right**).

## 4. Discussion

Initially, it was thought that the morphology of populations of *P. bimaculata* from the main channel Susquehanna River might differ from the populations inhabiting Octoraro Creek. No differences in the morphology of these populations were observed.

Furthermore, the populations of *P. c. caprodes* from the Allegheny River and *P. bimaculata* from the Susquehanna River are clearly different.

The Genesee River flows into Lake Ontario; thus, one might expect that they would be more similar to populations from Lake Erie. The Genessee River, however, has outlets to the Allegheny River via the Cuban glacial outlet [10,14], Oswayo Creek, and Honeoye Creek [15]. Connections between the Genesee and Susquehanna rivers included Cowanesque Creek and Pine Creek [16]. The fact that populations of *P. caprodes* from the upper Genesee River are more closely aligned morphologically with populations in the upper Allegheny River supports this connection. The Potomac River, where *P. bimaculata*, was native, heads up against the Youghiogheny River (Monongahela River tributary). The Savage River (Potomac River tributary) captured a portion of the Casselman River (Youghiogheny River tributary) [15–19]. Similarly, Gandy Creek (Monongahela tributary via the Cheat River) was captured by the North Fork of the South Branch of the Potomac River [20,21]. It should be noted that our specimens from the Genesee River were collected near Wellsville, New York, which is upriver of a series of waterfalls. These waterfalls have effectively isolated the upper river from fish invading from Lake Ontario.

Historically, *Percina nebulosa* and *P. bimaculata* were considered synonyms of *P. caprodes*. Near [6] elevated *P. bimaculata* to species status and because *P. nebulosa* was already occupied determined *P. bimaculata* to be the available and appropriate specific epithet. Thompson [22] showed the distribution of the populations of *P. caprodes* (*P. bimaculata*) in the Susquehanna and Potomac rivers as being completely disjunct from other populations of *P. caprodes* in Pennsylvania. Based on mtDNA and the S7 intron, Near [6] concluded that *P. bimaculata* was a sister species to a clade containing *Percina austroperca* and *Percina kathae*. *Percina kathae* is endemic to the Mobile Basin in Mississippi, Alabama, and Georgia [22], while *P. austroperca* is restricted to the Choctawhatchee and Escambia drainages in Alabama and Florida [23]. In 2011, however, Near [24] reported a phylogeny that aligned *P. bimaculata* more closely to *P. caprodes* than to either *P. austroperca* or *P. kathae*. It could not be determined if the *P. caprodes* reported by Near [25] were *P. c. caprodes* or *P. c. semifasciata*. Certainly, the native fish of the Susquehanna River system include those from the northern Mississippi River via the Allegheny River, along shorelines of glacial lakes in the developing Laurentian Basin [25], and ancient connections with the Finger Lakes that include *P. c. semifasciata* [16]. *Percina c. semifasciata* is known from the upper Mississippi River, the Great Lakes, and the Hudson Bay drainages [26]. There are also the Teays/Monongahela/Potomac connections (see [16]), which would have provided a route to the lower Greater Susquehanna channel during glacial periods. Furthermore, fish (e.g., *Enneacanthus gloriosus*) [27] may have invaded from the northeast and southeast. Although now probably extinct, *Etheostoma sellare* and *P. bimaculata* are the only recent (within the last 100 years) endemic fish on the Atlantic Slope that occur in and north of the Potomac and south of St. Lawrence. Although *P. bimaculata* and *P. c. semifasciata* are quite different morphologically (Figure 2), we postulate that *P. bimaculata* diverged from a population of *P. c. semifasciata* that originated in the Great Lakes and occupied the Potomac and Susquehanna rivers. Jenkins et al. [28] noted that both *Percopsis omiscomaycus* (Walbaum) and *P. c. semifasciata* (now *P. bimaculata*) reached the Potomac from the north since they are absent in the upper Monongahela system. They further state that a route using the Greater Susquehanna is favored by their occupation of large rivers. The close relationship of *P. c. semifasciata* and *P. bimaculata* is supported by the naked nape found in both of these species, the fact that both species do not occur above the fall line in Pennsylvania and may explain why Collette and Knapp [29] regarded *P. bimaculata* as a synonym of *P. c. semifasciata*. Knapp [30] considered scalation of the nape to be an important characteristic to separate the races of *Etheostoma caeruleum*, and Esmond and Stauffer [31] used morphological evidence to support their hypothesis that *E. caeruleum* was native to the Potomac River. Furthermore, most haplochromine cichlids are described on the basis of morphology alone because of the current lack of detectable fixed genetic differentiation [32,33].

## 5. Conclusions

A detailed morphological comparison of *P. bimaculata* with other species in the subgenus *Percina* is essential, given that there exists present or threatened destruction, modification, or curtailment of its habitat or range; the inadequacy of existing regulatory mechanisms; and other natural or manmade factors affecting its continued existence as listed in the Federal Register [34]. Efforts led by the Pennsylvania Fish and Boat Commission to reintroduce the Chesapeake logperch into its original habitat require that the populations that are introduced are representative of the races that were native. Certainly, a more detailed study, including molecular data on the systematics of the subgenus *Percina* is needed. It may be that *P. c. semifasciata* from the Great Lakes should be elevated to species status. The morphological and habitat (e.g., occurring below the fall line) data presented herein suggest that *P. bimaculata* is closely aligned with *P. c. semifasciata* from the Great Lakes.

**Author Contributions:** Conceptualization, J.R.S.J.; Data curation, J.R.S.J., J.A.F. and D.P.F.; Writing—original draft, J.R.S.J.; Writing—review & editing, J.A.F., D.P.F. and R.W.C.; Project administration, D.P.F. All authors have read and agreed to the published version of the manuscript.

**Funding:** Funding was provided by the Department of the Interior, the U.S. Fish and Wildlife Service, the Wildlife and Sport Fish Restoration Program, the Competitive State Wildlife Grant Program F18AS00095, and the USDA National Institute of Food and Agriculture, under Hatch project #PEN04582 (J.R.S.J.).

**Institutional Review Board Statement:** The study was approved by the Animal Use and Care Committee at Pennsylvania State University (protocol code PROTP201800659).

**Acknowledgments:** We appreciate the field assistance of Matt Ashton (Maryland Department of Natural Resources), Mitchell Bargo (PFBC), Kyle Clark (PFBC), Aaron Henning (Susquehanna River Basin Commission), Ben Lorson (PFBC) Drew Bucha and Antonios Stylianides, students from Penn State University, and Nathan Weyandt, a research assistant.

**Conflicts of Interest:** The authors declare no conflict of interest.

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
