# Peer review of "Morphological Comparison of the Chesapeake Logperch Percina bimaculata with the Logperch Percina c. caprodes and Percina c. semifasciata in Pennsylvania"

_fishes, doi:10.3390/fishes8060288_

Round 1

Reviewer 1 Report

The MS reports important results for a group of taxa whose taxonomy is still under debate. The 'confused' and not well knows taxonomy of a species represents one of the main threats to the survival of that species because this uncertainty can lead to the failure of adequate population management measures with consequences on the survival of the species itself. The authors report that some populations of the species examined have disappeared and have been subject to restocking activities. For this reason, this work represents an important point of reference to draw on for future management measures.

However, the article needs some editing before it will be considered. First of all, a map showing the stations where the samples were taken and also showing the distribution of the species would be very useful, especially for readers who are not confident about the places.

Another point to take under consideration is the lack of molecular data to support these findings. The debate about the validity of stand-alone morphological and stand-alone molecular studies is still open. For this, in the last years, the researchers are trying to merge both studies to support the taxonomical hypothesis. This approach gives more detailed findings and multifaced results. My concern here is why the authors decided to follow the ''old'' stand-alone morphological approach and do not decide to collect also data for a further molecular comparison among the studied taxa. 

Last, the discussion could be improved by reporting more examples of similar studies carried on for other taxa and using the same approach. Moreover, a paragraph at the end of the discussion to underline the limitation of the study and the need for further detailed research should be added.

Please find all the detailed comments in the pdf attached. 

Author Response

Reviewer 1

I used extirpated because after several attempts, the Maryland DEP could not find it in the Potomac drainage and it is listed as extirpated.

Removed All of he above are members of the subgenus Percina (line 53)

Included a map showing the lotic habitats in PA and MD where collections were made.

Line 79-90 – the collection sites are not repeated in the results. Since it lists where they were collected it should be part of the methods

Lines 179-185 were removed.

Corrected spelling of semifasciata ln 216

Removed spaces ln 221

By recent I mean in our life time. There may have historically been other endemic species. I have added in the last 100 years.

Cited other works which relied on morphology.

Added a paragraph stating what needs to be completed in the future.

Reviewer 2 Report

The manuscript under review is devoted to a rather deep morphological study of Percina bimaculata and comparison with two subspecies of another species of the same subgenus, living in Pennsylvania and Maryland. The work was carried out by classical morphological methods accepted in ichthyology, at the same time in the discussion, the authors draw on the literature with DNA analyzes, which seems to be absolutely necessary for modern studies of this kind. The significance of Percina bimaculata from the point of view of fisheries is not great, but given its status in the Red Book, a detailed study, including morphological and biogeographical areas, seems to be quite justified. 

Both in terms of the object of study and the quality of processing of the material, it can be recommended for publication in a specialized journal for the study of fish, such as Fishes. The literature review, materials and methods, the results are presented very professionally and at a level sufficient for understanding.

But the Discussion section, in my opinion, needs to be improved. The main remarks are as follows: the text lacks both structurally and in meaning the concluding part (Conclusions), which is absolutely necessary in a comparative study of such a very heterogeneous material.

I would also like to understand, and therefore it should be included in the description of the work and in conclusion, a clearer research objective. If authors write "to morphologically compare populations of this subgenus that inhabit Maryland and Pennsylvania", then it is necessary to indicate what significance this task could play in the framework of a more relevant and more understandable for readers task related to the Red Book status of this species. Since the species is clearly in need of protection, it would be very useful if the results obtained were somehow correlated with recommendations for recording/monitoring/conservation of this species within the natural range. 

Therefore, I strongly recommend linking the research objectives with the Red Book status of this species.

Author Response

Reviewer 2

Added the status of the Chesapeake Logperch relative to its existence. Added conclusion and noted that reintroduction requires that the same populations be translocated.